# Downregulation of Salt-Inducible Kinase 3 Enhances CCL24 Activation in the Placental Environment with Preeclampsia

**DOI:** 10.3390/ijms25010222

**Published:** 2023-12-22

**Authors:** Hsing-Fen Tsai, Ching-Fen Tseng, Yu-Ling Liang, Pei-Ying Wu, Lan-Yin Huang, Yu-Han Lin, Li-Hsuan Lin, Chang-Ni Lin, Keng-Fu Hsu

**Affiliations:** 1Department of Obstetrics and Gynecology, National Cheng Kung University Hospital, College of Medicine, National Cheng Kung University, Tainan 704302, Taiwan; pockey1202@gmail.com (H.-F.T.); ulin945@gmail.com (Y.-L.L.); anna1002ster@gmail.com (P.-Y.W.); lanyin0801@gmail.com (L.-Y.H.); a0201a0201@hotmail.com (Y.-H.L.); 2Department of Biochemistry and Molecular Biology, National Cheng Kung University, Tainan 70101, Taiwan; j3eanita@gmail.com; 3Department of Obstetrics and Gynecology, Tainan Hospital, Ministry of Health and Welfare of Taiwan, Tainan 70101, Taiwan; dramaqueen0019@hotmail.com; 4Institute of Clinical Medicine, College of Medicine, National Cheng Kung University, Tainan 70101, Taiwan

**Keywords:** preeclampsia, SIK3, CCL24, M2 skewing, IL-10

## Abstract

Preeclampsia (PE) remains one of the leading causes of maternal and perinatal morbidity and mortality. However, the exact pathophysiology of PE is still unclear. The recent widely accepted notion that successful pregnancy relies on maternal immunological adaptation is of utmost importance. Moreover, salt-inducible kinase 3 (SIK3) is an AMP-activated protein kinase-related kinase, and it has reported a novel regulator of energy and inflammation, and its expression related with some diseases. To explore whether SIK3 expression correlated with PE, we analyzed SIK3 gene expression and its association with PE through GEO datasets. We identified that SIK3 was significantly downregulated in PE across four datasets (*p* < 0.05), suggesting that SIK3 participated in the pathogenesis of PE. We initially demonstrated the significant downregulation of SIK3 in trophoblast cells of PE. SIK3 downregulation was positively correlated with the increased number of CD204(+) cells in in vivo and in vitro experiments. The increased number of CD204(+) cells could inhibit the migration and invasion of trophoblast cells. We then clarified the potential mechanism of PE with SIK3 downregulation: M2 skewing was triggered by trophoblast cells derived via the CCL24/CCR3 axis, leading to an increase in CD204(+) cells, a decrease in phagocytosis, and the production of IL-10 at the maternal–fetal interface of the placenta with PE. IL-10 further contributed to a reduction in the migration and invasion of trophoblast cells. It also established a feedback loop wherein trophoblast cells increased CCL24 production to maintain M2 dominance in the placental environments of PE.

## 1. Introduction

Preeclampsia (PE), a leading contributor to maternal and perinatal morbidity and mortality, is associated with incomplete uterine vascular remodeling, leading to shallow trophoblast invasion. This condition results in intrauterine growth restriction (IUGR) and maternal syndromes of hypertension, proteinuria, and/or end-organ damage, which usually manifest after the mid-trimester of pregnancy [1,2,3,4]. The combined biomarkers of PE, including cytokines, proteins, and angiogenic and antiangiogenic factors, play pivotal roles in the pathogenesis and etiology of PE, as well as in other related hypertensive disorders of pregnancy. However, the exact pathophysiology of PE remains unclear, and the early prediction of PE acts as a significant challenge.

A successful pregnancy depending on maternal immunologic tolerance systems to balance the allogeneic fetus and placenta is wildly accepted now [5]. Conversely, the pathogenesis of PE is related to inadequate spiral artery remodeling and poor trophoblast invasion; consequently, it creates hypoxic placental environments and causes maternal systemic inflammation disorders [6,7]. In other words, an elevated inflammatory load in pregnant women plays an important role in the pathogenesis of PE. Interestingly, studies have shown that the inhibition of SIK2 and SIK3 during macrophage differentiation significantly increases the production of IL-10, an anti-inflammatory cytokine [8]. In our previous study, we found that knocking down the *salt-inducible kinase 3* (*SIK3*) gene in an ovarian cancer cell line triggers the upregulation of IL-1β and TNF-α genes and increases the release of proinflammatory cytokines [9]. Furthermore, the increased levels of TNF-α and IL-6 in the circulation of preeclamptic women have been reported, suggesting that factors beyond the placenta influence these levels [10]. In other words, the detected levels of TNF-α or IL-6 in circulation cannot serve as reliable predicable markers of PE. To date, increasing evidence suggests that an imbalance in immunologic modulation at the maternal–fetal interface of the placenta contributes to certain pregnancy complications, such as PE and IUGR [11]. Given the dynamic function of SIK3 in immune reaction, we hypothesized that SIK3 may play a role in the pathogenesis of PE, especially in macrophage differentiation.

The decidua, the maternal–fetal interface of the placenta, can regulate trophoblast cells invasion through maternal immune cells and their secreted products within the microenvironments [12]. Accumulating evidence indicates that the decidua contains various immune cells, including decidual natural killer cells (dNK), decidual macrophages (dM), T cells, and to a lesser extent, dendritic cells (DC) and B cells, creating a unique environment for the maternal immune system to develop tolerance toward fetal antigens [13,14,15]. The abnormal function and phenotype of dM have been implicated in disrupting immune tolerance through the regulation of genes such as *CXCL3*, *ICAM1*, *IGF1*, *CXCL12*, *NRP1*, and *LRP6*, ultimately resulting in PE [16]. In the present study, we aimed to explore the role of SIK3 in placental environments and its involvement in altering dM phenotype and function. We also aimed to propose the potential mechanism of PE.

## 2. Results

### 2.1. SIK3 Downregulation in Trophoblasts Is Associated with an Increase in CD204(+) Cells in the Decidua

To investigate the role of SIK3 in PE, we first analyzed the SIK3 gene expression in the placenta sample with/without PE in the GEO datasets. Seven datasets, namely GSE4707, GSE10588, GSE14722, GSE24129, GSE30186, GSE43942, and GSE54618, were included. GSE4707, GSE24129, GSE30186, and GSE54618 revealed a significant reduction in *SIK3* in the placenta with PE compared with that in normal pregnancies (Table 1). Afterward, we performed immunohistochemistry staining on 14 placenta samples (with/without PE). The immunohistochemistry (IHC) staining results revealed a significant downregulation of SIK3 in PE (N = 7) compared with that in non-PE (N = 7; *p* = 0.0008; Figure 1A,B). The number of CD204(+) cells, not CD68(+) cells, notably increased in the decidua with PE compared with that in the decidua with non-PE (*p* = 0.0014; Figure 1C,D). Importantly, the SIK3 expression was significantly negatively correlated with the number of CD204(+) cells (r^2^ = 0.3103, *p* = 0.0385; Figure 1E). Statistical analysis revealed that gestational age (mean: 39.4 ± 0.7 weeks vs. 36.2 ± 2.4 weeks, *p* = 0.018), birth weight of babies (mean: 3129.3 ± 244.0 g vs. 2257.7 ± 698.2 g, *p* = 0.005), SIK3 expression (*p* = 0.008), and CD204 expression (*p* = 0.008) were substantially different between non-PE and PE (Table 2).

### 2.2. Low SIK3 Expression at the Maternal–Fetal Interface of the Placenta Contributed to M2 Skewing and Reduced Phagocytosis in the In Vitro Model

We used shRNA to knock down SIK3 in HTR-8 and 3A-sub E cell lines to explore the interaction between SIK3 downregulation and the increase in CD204(+) cells. We observed that a high SIK3 expression exclusively in the 3A-sub E cell line was effectively knocked down by shRNA (Figure 2A). Then, we cocultured the *SIK3*-shRNA 3A-sub E cells with the monocytic cell line (THP-1) to simulate the placenta microenvironments of PE, representing the maternal–fetal interface of the placenta (Figure 2B). After 7-day coculture, we examined the population of CD204(+) cells in THP-1 cells through immunofluorescence staining. Flow cytometry results showed an impressive increase in the CD204(+) cell population after the coculture with *SIK3*-shRNA 3A-sub E#01 (mean = 54.62 ± 2.96%, *p* = 0.0003) or *SIK3*-shRNA 3A-sub E#61 (mean = 48.08 ± 4.11%, *p* = 0.0061) compared with that after the coculture with SIK3-shRNA 3A-sub E#Luc (mean = 29.29 ± 3.54%; Figure 2C). The phagocytic ability in THP-1 cells decreased by approximately 50% after the coculture with *SIK3*-shRNA 3A-sub E#01 (mean: 25.98%, *p* = 0.0022) and approximately 30% after the coculture with *SIK3*-shRNA 3A-sub E#61 (mean: 34.32%, *p* = 0.0228) compared with that after the coculture with *SIK3*-shRNA 3A-sub E#Luc (mean: 45.13%; Figure 2D). These results suggested that the low SIK3 expression at the maternal–fetal interface of the placenta was involved in M2 macrophage skewing in pregnant women with PE.

### 2.3. Low SIK3 Expression at the Maternal–Fetal Interface of the Placenta Reduced the Invasion and Migration Ability of Trophoblast Cells

Considering that the decidua could affect trophoblast cell invasion via cytokine production, we demonstrated that the low SIK3 expression at the maternal–fetal interface of the placenta contributed to M2 skewing in the coculture model. Next, we investigated whether the invasion and migration of trophoblast cells with a low SIK3 expression changed at the maternal–fetal interface of the placenta. We treated *SIK3*-shRNA 3A-sub E cells with the condition medium (CM) obtained from 3A-sub E cocultured with THP-1 cells for 7 days and analyzed the cell invasion/migration ability. The cell invasion abilities of *SIK3*-shRNA 3A-sub E#01 (0.60 fold, *p* = 0.0358) and SIK3-shRNA 3A-sub E#61 (0.56 fold, *p* = 0.0312) cells significantly decreased after CM treatment compared with those after the treatment with RPMI 1640 medium (fold = 1.00; Figure 3A). The same results were observed in the wound healing assay (Figure 3B). These findings implied that some factors present in the CM could profoundly affect the invasion and migration of trophoblast cells.

### 2.4. Trophoblast-Derived CCL24 Positively Affected Macrophage Recruitment at the Maternal–Fetal Interface of the Placenta and Enhanced M2 Skewing

To understand the crucial factor that is abundant in the placental environments and its association with PE, we conducted cytokine array analysis by using six cord blood samples from the umbilical vein with/without PE. The cytokine array data showed that the levels of CCL24, CCL26, FGF6, IGF-1, IL-10, and IL-15 in the plasma samples with PE (3 samples mixed) were higher than those without PE (3 mixed samples; Figure 4A). Among these cytokines, CCL24 exhibited the highest concentration in the cord blood with PE compared with non-PE. We then used a CCL24 ELISA kit to validate the CCL24 concentration in the CM and in 14 cord blood samples with/without PE. After the coculture with THP-1 cells for 7 days, the CCL24 concentration increased significantly in the CM of *SIK3*-shRNA 3A-sub E#01 (41.33 ± 6.67 pg/mL, *p* = 0.0074) and *SIK3*-shRNA 3A-sub E#61 (34.67 ± 11.55 pg/mL, *p* = 0.0150) compared with the CM of *SIK3*-shRNA 3A-sub E#Luc (5.33 ± 2.67 pg/mL), respectively (Figure 4B, red histogram). Conversely, the CCL24 concentration revealed no significant change in the CM of *SIK3*-shRNA 3A-sub E without the coculture with THP-1 cells (Figure 4B, blue histogram). Similarly, the CCL24 concentration in the cord blood samples with PE was higher than that in the non-PE samples (157.14 ± 31.16 pg/mL vs. 54.29 ± 3.39 pg/mL, *p* = 0.0066; Figure 4C).

To identify the source of CCL24, we analyzed the *CCL24* transcription level in *SIK3*-shRNA 3A-sub E and THP-1 cells. The qPCR data demonstrated a remarkable increase in the *CCL24* transcription levels in *SIK3*-shRNA 3A-sub E#01 (13.04 fold, *p* = 0.0134) and *SIK3*-shRNA 3A-sub E#61 (4.99 fold, *p* = 0.0499) after the coculture with THP-1 cells for 7 days. By contrast, the *CCL24* transcription level remained unchanged in THP-1 cells whether they were cocultured with or without SIK3-shRNA 3A-sub E cells (Figure 4D). These findings implied that the source of CCL24 was trophoblast cells with SIK3 downregulation, and CCL24 was abundantly increasing in the placental environments with PE.

SIK3 downregulation was associated with the increasing number of CD204(+) cells, whereas the downstream of SIK3 that we demonstrated here was CCL24. We further investigated the role of CCL24 in M2 skewing in the placental environments with PE. Flow cytometry revealed a 25% increase in the CD204(+) population in THP-1 cells after the treatment with 0.05 ng/mL recombinant CCL24 (mean = 36.32 ± 7.40%) compared with that in the untreated cells (mean = 11.39 ± 4.07%, *p* = 0.0419; Figure 5A). The CM treatment (positive control) also demonstrated a 31% significant increase (*p* = 0.0081). Chemoattractant results showed that the migrated THP-1 cells significantly increased after the CM (2.34-fold, *p* = 0.0006) or 0.05 ng/mL CCL24 treatment (2.27-fold, *p* = 0.0028; Figure 5B). Importantly, CCL24 and CCR3 were obviously co-expressed at the maternal–fetal interface of the placenta (Figure 5C), while CCR3 is the specific receptor for CCL24 [17].

We further used SB 297006, a CCR3 antagonist, to validate the CCL24/CCR3 axis whether is the major contributor to M2 polarization in the placental environments with PE. We observed that the CD204(+) cell population decreased significantly in THP-1 cells after the coculture with *SIK3*-shRNA 3A-sub E#01 and 2.5 μM SB 297006 (57.98 ± 5.42% reduction to 36.78 ± 2.57%, *p* = 0.0241). The same result was observed after the coculture with *SIK3*-shRNA 3A-sub E#61 and treatment with 2.5 μM SB 297006 (54.96 ± 5.69% reduction to 37.77 ± 1.83%, *p* = 0.0452) (Figure 5D). Additionally, the SB 297006 exposure could reverse the phagocytotic effect on THP-1 cells, regardless coculture with *SIK3*-shRNA 3A-sub E#01 (21.09 ± 2.61 increased to 43.81 ± 2.88, *p* = 0.0043) or *SIK3*-shRNA 3A-sub E#61 (29.45 ± 2.30 increase to 41.34 ± 2.16, *p* = 0.0197) (Figure 5E). These results suggested that trophoblast cell-derived CCL24 sufficiently altered the phenotype and function of maternal macrophages in the placental environments with PE by interacting with CCR3.

### 2.5. CCL24/CCR3 Axis Played a Pivotal Role in Polarizing Maternal Macrophages to Produce IL-10, Which Effectively Inhibited Trophoblast Cell Invasion/Migration

To understand the roles of these increasing M2 cells in the placental environments with PE, we observed that IL-10 is another abundant cytokine in cord blood samples with PE. Herein, we analyzed the source of IL-10 in in vitro experiments. qPCR data showed that the transcription level of *IL-10* significantly increased in THP-1 cells after the coculture with *SIK3*-shRNA 3A-sub E#01 (8.37-fold, *p* = 0.0004) or after the coculture with *SIK3*-shRNA 3A-sub E#61(2.49-fold, *p* = 0.0208; Figure 6A). ELISA revealed that the IL-10 concentration in the cord blood samples of PE (mean: 8.17 pg/mL) was significantly higher than that in the cord blood samples of non-PE (mean: 3.70 pg/mL, *p* = 0.0484; Figure 6B). After the coculture with THP-1 cells, a higher IL-10 concentration was detected in *SIK3*-shRNA 3A-sub E#01 CM (mean: 6.13 pg/mL, *p* = 0.0214) and the *SIK3*-shRNA 3A-sub E#61 CM (mean: 5.65 pg/mL, *p* = 0.0080) than in the coculture with *SIK3*-shRNA 3A-sub E#Luc (mean: 0.80 pg/mL; Figure 6C). In the CM of THP-1 cells cocultured with *SIK3*-shRNA 3A-sub E#01, the level of IL-10 could be significantly reduced upon 2.5 μM SB 297006 exposure (7.88 ± 0.36 pg/mL reduced to 2.25 ± 0.36 pg/mL, *p* = 0.0004). The same result can be observed in the CM of THP-1 cells cocultured with *SIK3*-shRNA 3A-sub E#61 after 2.5 μM SB 297006 treatment (6.00 ± 0.63 pg/mL reduced to 2.67 ± 0.55 pg/mL, *p* = 0.0161) (Figure 6D).

After recombinant IL-10 (8 pg/mL) treatment, more CCL24 was detected in *SIK3*-shRNA 3A-sub E#01 CM (mean: 11.33 pg/mL, *p* = 0.0215) and *SIK3*-shRNA 3A-sub E#61 CM (mean: 7.44 pg/mL, *p* = 0.1056) compared with that in the untreated sample (Figure 6E). The cell invasion of *SIK3*-shRNA 3A-sub E#01 (0.75-fold, *p* = 0.0053) and *SIK3*-shRNA 3A-sub E#61 (0.80-fold, *p* = 0.0040) was significantly reduced by 8 pg/mL recombinant IL-10 treatment compared with that of *SIK3*-shRNA 3A-sub E#Luc (1.00-fold; Figure 6F), whereas the migration of *SIK3*-shRNA 3A-sub E#01 (0.90-fold, *p* = 0.2053) and *SIK3*-shRNA 3A-sub E#61 (1.06-fold, *p* = 0.3217) was not affected (Figure 6G). These findings suggested that the downregulation of SIK3 in trophoblast cells contributed to CCL24 secretion to enhance M2 skewing at the maternal–fetal interface of the placenta via the CCL24/CCR3 axis, resulting in an increase in CD204(+) cells, a decrease in phagocytosis, and production of abundant IL-10. These IL-10 could inhibit trophoblast cell invasion/migration and prompt trophoblast cells to induce more CCL24 production to maintain the M2 dominance in the placental environment of PE. However, blocking the interaction of CCL24 and CCR3 could reverse the M2 dominance in the placental environment of PE. The schematic of the placental environment with PE was showed on Figure 7.

## 3. Discussion

Our findings revealed that the SIK3 expression at the maternal–fetal interface of the placenta was lower in women with PE than in women with normal pregnancy, and the number of macrophages at the maternal–fetal interface of the placenta was M2-phenotype dominant. SIK3 serves as a novel regulator of energy and inflammation, playing a dynamic role in immune modulation [6,7]. A comprehensive analysis of SIK3 gene expression in normal pregnancy and PE datasets indicated a significant reduction in four out of seven (4/7) datasets (Table 1). The pathogenesis of PE is associated with inadequate spiral artery remodeling and poor trophoblast invasion, leading to a hypoxic placental environment and maternal systemic inflammation disorders [8,9]. However, Benyo et al. reported that the concentrations of the proinflammatory factors TNF-alpha and IL-6 show no significant difference in the circulation of preeclamptic women, suggesting that sources other than the placenta contribute to the elevated concentrations of proinflammatory cytokines [10]. This result illustrated that circulating cytokine detection is not suitable for preeclampsia diagnosis. We demonstrated that a low SIK3 expression at the maternal–fetal interface of the placenta was involved in M2 macrophage skewing, resulting in changes in the placental environment of pregnant women with PE. Therefore, circulating SIK3 detection might act as a preeclampsia prediction marker.

Our study revealed that the pregnant women with PE delivered at an earlier gestational age (36.2 ± 2.4 weeks). At the maternal–fetal interface of the placenta with PE, a lower SIK3 expression and a higher CD204 (as M2 macrophage marker) expression were observed (Table 2). During early pregnancy, macrophages comprise approximately 20–30% of decidual leukocytes and contribute to spiral artery remodeling by producing angiogenic factors [18]. The macrophage population includes proinflammatory and anti-inflammatory macrophages (M1 and M2 macrophages). M1 macrophages dominate in the first trimester (<12 weeks of gestation) and contribute to embryo implantation and placental formation. Furthermore, more macrophages are polarized into M2 macrophages to maintain maternal–fetal tolerance in the second trimester; at term pregnancy, decidual macrophages are involved in initiating labor, and M1 macrophages become dominant again [19,20]. This mechanism implies that at the maternal–fetal interface, the number and proportion of M1/M2 macrophages change during different gestational ages to protect the fetus from the maternal immune microenvironment and tolerate the allograft (fetus). In our study, the participants, including those in the group with PE, were in their third trimester and at near-term pregnancy. However, the expression of the M2 macrophage marker was higher at the maternal–fetal interface of the placenta with PE; this finding was contrary to previous studies, which showed that the decidual macrophage phenotype exhibits M1 dominance in pregnant women with severe PE [21,22]. We inferred that the lower SIK3 expression at the maternal–fetal interface of the placenta with PE changed the placental environment and contributed to the phenotypic skewing of the M2 macrophage, which established an anti-inflammatory immune tolerance system in the placenta with PE.

Even though the detection of circulating cytokines as prediction markers is under debate, they play pivotal roles in immune systems with PE. The imbalance between proinflammatory and anti-inflammatory cytokines leads to the pathophysiology associated with PE [23]. In patients with PE, the imbalance of cytokines results in chronic systemic inflammation and local placental inflammation. In our study, cytokine array data showed a marked increase in the CCL24 level in the PE cord blood (Figure 4). CCL24 promotes cell growth and induces M2 macrophage phenotype shifting by interacting with a unique receptor, i.e., CCR3 [19,24,25]. CCL24 is essential for cell proliferation and invasion in early pregnancy [24,25,26]. CCR3 is primarily expressed on the membrane of eosinophils, basophils, monocyte-derived dendritic cells, and a subset of Th2 lymphocytes involved in allergy and immune-mediated inflammation [27]. In our in vitro experiments, we observed that THP-1 cell migration significantly increased with rhCCL24 and CM treatment (Figure 5B); immunofluorescence data also indicated a significantly higher co-expression of CCR3 and CD204 at the maternal–fetal interface of the placenta with PE than those without PE (Figure 5C). These results implied that CCL24 acted as a chemoattractant for macrophages at the maternal–fetal interface and contributed to M2 skewing by interacting with CCR3.

Interestingly, we noticed an increase in the IL-10 level in the cord blood of PE through cytokine array analysis (Figure 4A), while IL-10 has been demonstrated as a key anti-inflammatory mediator to regulate wound healing, autoimmunity, cancer, and homeostasis [28]. The dysregulation of IL-10 is associated with the occurrence of PE, although the IL-10 detection yields inconsistent results [29]. Here, we demonstrated that the increased IL-10 in the placental environments with PE was from M2 macrophages (Figure 6). The increased IL-10 contributed to a reduction in trophoblast cell migration/invasion (Figure 6) and enhanced the production of CCL24 (Figure 6E). Our findings differed from most studies suggesting a decrease in circulating IL-10 in pregnant women with PE [29,30,31]. This variance could be attributed to our use of cord blood from the umbilical vein to analyze IL-10 level, which is more representative for the placental environments with PE. However, the majority of the pregnant women with PE in our study were in the late preterm gestational age (36.2 ± 2.4 weeks), and only two cases had severe features. Our results presented the environments of preeclamptic placenta with less severity. This novel finding highlighted that the regulation of IL-10 in the placental environments with PE was mediated by the decreased expression of SIK3 in trophoblast cells of PE.

This study highlighted the significant role of SIK3 downregulation in activating CCL24 in trophoblast cells, leading to an increased level of CD204(+) cells at the maternal–fetal interface of the placenta with PE. IL-10 released from these CD204(+) cells served to reduce the migration/invasion of trophoblast cells and to produce more CCL24 from trophoblast cells to maintain M2 dominance in the placental environments with PE. The findings suggested that SIK3 could be a potential marker for future PE prediction, and further exploration of the role of CCL24 in PE is warranted.

## 4. Materials and Methods

### 4.1. Clinical Sample Collection

Placenta samples with and without PE were collected from January 2022 to December 2022 after the newborns’ births. With the umbilical cord as the center, approximately 3 × 3 × 3 cm^3^ rectangular prism was cut from the amniotic membrane surface to the decidua, and approximately 1 × 1 × 1 cm^3^ cubes were cut near the decidua to perform tissue embedding. The study protocol was approved by the Institutional Review Board of NCKUH (No: B-ER-110-314), and informed consent was obtained from all pregnant women. Clinical information, including the age of pregnant women, gestational age, and newborn’s weight, was collected to assess clinical inflammation.

### 4.2. Immunochemistry Staining

Formalin-fixed paraffin-embedded tissue samples were obtained and stained with anti-human CK7 antibody (1:500 dilution; ab183344; Abcam, Inc., Cambridge, UK) as the positive control for trophoblast cells. Anti-human SIK3 antibody (1:100 dilution; Dr. provided by Neng-Yao Shih), anti-human CD68 antibody (1:500 dilution; M0814; Agilent Technologies, Inc., Santa Clara, CA, USA), or anti-human CD204 antibody (1:200 dilution; MAB1710; Abnova Corporation) was used to stain the trophoblast cells and macrophages near the decidua. The sections were serially dewaxed, rehydrated, and heated with 10 mM sodium citrate (pH 6.0) for 20 min for antigen retrieval. After endogenous peroxidases were blocked with 3% hydrogen peroxide, the sections were incubated with the primary antibody overnight at 4 °C. Horseradish peroxidase (HRP)-conjugated immunoglobulin G (IgG) antibody was added and incubated for 1 h, and the specimens were analyzed through avidin and biotinylated enzyme complex (ABC) detection. Images were obtained under a ZEISS Axio Imager D2 microscope, and quantification (%Area) based on three 200× fields was performed using Nuance imaging software (version 3.0.2, PerkinElmer, Burlington, VT, USA).

### 4.3. Cell Culture and Transfection

The human trophoblast cell line 3A-sub E was cultured in RPMI 1640 medium (Thermo Fisher Scientific, Waltham, MA, USA) supplemented with 10% fetal bovine serum (HyClone, GE Healthcare Life Sciences, South Logan, UT, USA) and 0.1 mM sodium pyruvate. THP-1, a human monocytic cell line, was cultured in RPMI 1640 medium (Thermo Fisher Scientific, Waltham, MA, USA) supplemented with 10% fetal bovine serum (HyClone, GE Healthcare Life Sciences, South Logan, UT, USA). The *SIK3* lentiviral expression plasmid (RNAi Core of Academia Sinica) and packaging vectors (ViraPowerTM Packaging Mix; Invitrogen, Waltham, MA, USA) were used to silence the gene expression of *SIK3* in 3A-sub E cells. Next, the 3A-sub E cells with and without *SIK3* knockdown were cocultured with THP-1 cells by using a Boyden chamber with a 0.4 nm pore size insert to simulate the placental environment with and without PE. After a 7-day incubation period, the cell lysates and culture medium were collected and stored at −20 °C until further analysis. All cells were incubated in a humidified atmosphere at 37 °C with 5% CO_2_.

### 4.4. Western Blot

In this procedure, 50 μg of protein was subjected to 10% SDS-PAGE and subsequently transferred onto PVDF membranes (IPVH00010; Merk Millipore, Burlington, MA, USA). The mouse anti-human SIK3 (1:500 dilution; provided by Dr. Neng-Yao Shih) was applied overnight at 4 °C. After the primary antibody was removed and the PVDF membranes were washed, an anti-mouse HRP conjugate antibody (1:5000 dilution; 211-035-109; Jackson ImmunoResearch Laboratories, Inc., Pennsylvania, PA, USA) was added at room temperature for 1 h. The blocking/dilution reagent was composed of 5% skim milk in TBST (TBS plus 0.05% Tween-20). Proteins were visualized using an enhanced chemiluminescence system (Merk Millipore, Burlington, MA, USA) and detected with an X-ray film (Cytiva Amersham™ Hyperfilm™ MP; Thermo Fisher Scientific Inc., Waltham, MA, USA).

### 4.5. Phagocytosis Effect Detection

pHrodo^TM^ BioParticles^®^ Conjugate for Phagocytosis kit (Cat. P35361, Invitrogen, Inc. Waltham, MA, USA) was utilized to determine the phagocytic ability of THP-1 cells. After 7 days of coculture or without coculture, 1 × 10^6^ cells were seeded in a 6 cm dish, and 200 μg/100 μL pHrodo^TM^ BioParticles^®^ conjugates were added and incubation at 37 °C for 2 h. As a negative control, cytochalasin D (20 μM; C8273; Merk Millipore, Burlington, MA, USA), an inhibitor of actin polymerization, was preincubated with cells for 30 min before beads were added. The phagocytosis effect of THP-1 cells was detected via flow cytometry (FACS; BD Biosciences, San Jose, CA, USA).

### 4.6. Flow Cytometry

THP-1 cells were stained with fluorescein isothiocyanate (FITC)-conjugated anti-human CD204 (MAB1710; Abnova Corporation, Taipei, Taiwan) to analyze the quantity of M2-type macrophages after the coculture with *SIK3*-shRNA 3A-sub E; a fluorescence-activated cell sorter (FACS; BD Biosciences) was used for further analyses. The geometric mean fluorescence intensity (MFI) was determined with the FCSalyzer software (https://sourceforge.net/projects/fcsalyzer/, accessed on 19 December 2023).

### 4.7. Invasion/Migration Assay

A total of 5 × 10^5^ cells were seeded into an 8 μm pore polycarbonate insert with or without Matrigel (354234; Corning Inc. New York, NY, USA) and incubated at 37 °C. After 24 h, the membrane was removed carefully, washed in PBS, fixed in 4% paraformaldehyde, and stained with Giemsa (GS-10; Sigma-Aldrich, Inc., St Louis, MO, USA). These invasive/migrated cells were captured and quantified via light microscopy.

### 4.8. Cytokine Array

Human Cytokine Antibody Array (ab169817; Abcam, Cambridge, UK), a semiquantitative protein profiling method, was used to screen differentially expressed cytokines in the plasma of patients with PE compared with those with a normal pregnancy. The multiplex protein detection kit contained four membranes, and each membrane was spotted in duplicate with 42 different cytokine antibodies. The membranes were incubated with 1 mL of the plasma at 4 °C overnight in accordance with the manufacturer’s instructions. After incubation with a detection antibody cocktail, antibody conjugation, and recommended washes, the immunoblots on the membrane were developed with a chemiluminescent substrate reagent and exposed to an X-ray film. Cytokine array TIFF file images were analyzed using ImageJ software (https://imagej.en.download.it/, accessed on 19 December 2023). Normalization with a positive control spot was performed using ImageJ software after the intensity of every spot was obtained. The fold change of cytokines in the subjects with PE was calculated and compared with that in the subjects with normal pregnancy.

### 4.9. ELISA

CCL24 and IL-10 expression levels in the condition medium (CM) and the cord blood from the umbilical vein were measured using a commercially available ELISA kit (DCC240B, DY217B; R&D Systems Inc., Minneapolis, MN, USA) in accordance with the manufacturer’s instructions.

### 4.10. Real-Time PCR

An RNA extraction kit (740955.50; MACHEREY-NAGEL GmbH & Co. KG, Düren, Germany) was used in accordance with the manufacturer’s instructions. cDNAs were synthesized using the ImPromII^TM^ Reverse Transcription System (A3800; Promega, Madison, WI, USA). PCR was conducted on a StepOnePlus Real Time PCR System (Applied Biosystems, Foster City, CA, USA). Triplicate mean values were calculated using GAPDH gene transcription as the reference for normalization. The following primer sequences were used in qPCR analysis: CCL24, 5′-TGAGAACCGAGTGGTCAGCTAC-3′ (forward) and 5′-TTCTGCTTGGCGTCCAGGTTCT-3′ (reverse); and GAPDH, 5′-GGCTGAGAACGGGAAGCTTG-3′ (forward) and 5′-ATCCTAGTTGCCTCCCCAAA-3′ (reverse).

### 4.11. Statistical Analysis

Data were statistically analyzed using GraphPad Prism 8 software (La Jolla, CA, USA). Statistical significance was set at *p* < 0.05. Data from three independent studies were presented as mean ± standard error (SE). Statistical differences between means were analyzed through a paired Student’s *t*-test.

## Figures and Tables

**Figure 1 ijms-25-00222-f001:**
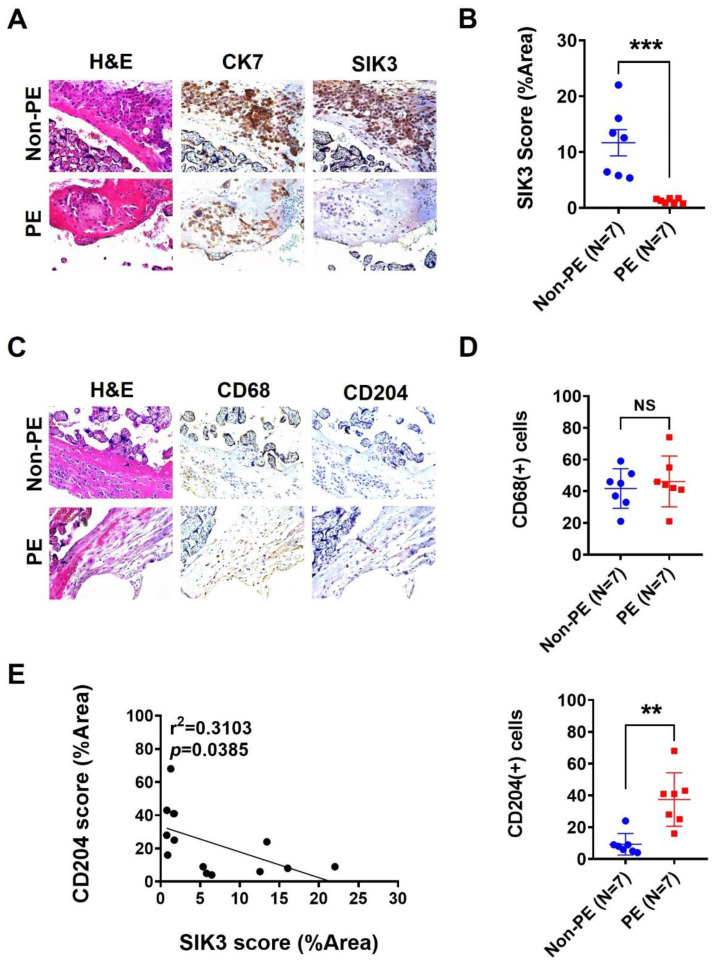
Low SIK3 expression in trophoblast cells with preeclampsia (PE) was correlated with M2-type macrophage accumulation at the maternal–fetal interface with placenta. (**A**,**B**) Seven placenta specimens of non-PE and seven placenta specimens of PE were subjected to immunohistochemistry staining. The SIK3 expression was significantly decreased in the placenta specimens of PE compared with that in the placenta specimens of non-PE (*p* = 0.0097). (**C**,**D**) After CD68 (as a pan-macrophage marker) and CD204 (as an M2 macrophage marker) staining, CD204-positive cells significantly increased at the maternal–fetal interface of the placenta with PE compared with those of the placenta with non-PE (*p* = 0.0094); however, the total number of macrophages did not significantly change (*p* = 0.3066). (**E**) The SIK3 expression was significantly negatively correlated with the CD204 expression in the placenta (r^2^ = 0.2908, *p* = 0.0466). ** *p* < 0.01; *** *p* < 0.001. NS: not significant.

**Figure 2 ijms-25-00222-f002:**
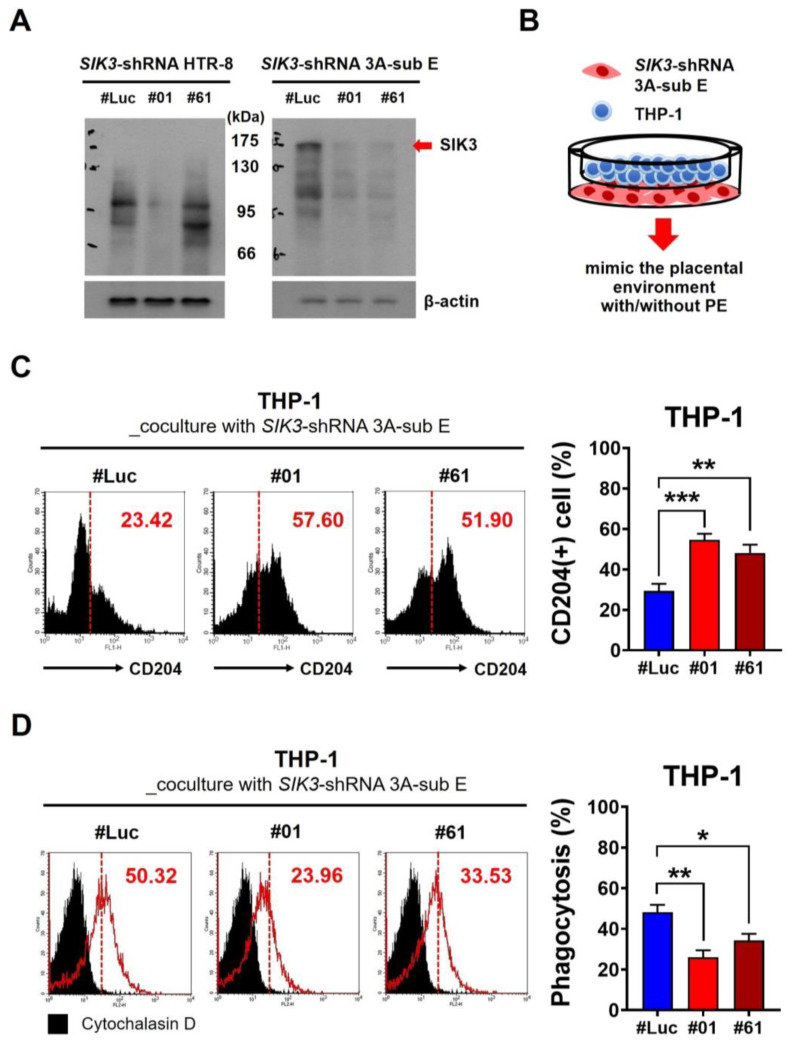
Downregulation of SIK3 in trophoblast cells contributed to the phenotypic skewing of the M2 macrophage and a reduction in phagocytic function. (**A**) Short hairpin RNA was used to downregulate the SIK3 expression in the human trophoblast cell line 3A-sub E (named *SIK3*-shRNA 3A-sub E) for further analysis. HTR-8, another trophoblast cell line, lacks SIK3 expression. (**B**) The *SIK3*-shRNA 3A-sub E#01 or SIK3-shRNA 3A-sub E#61 cells cocultured with THP-1 cells (human monocyte cell line) were used to establish an in vitro model of PE, whereas *SIK3*-shRNA 3A-sub E#Luc was cocultured with THP-1 as the in vitro model of non-PE. (**C**) In THP-1 cells, the CD204-positive cells significantly increased after the coculture with *SIK3*-shRNA 3A-sub E#01 (57.60%, *p* = 0.0003) or *SIK3*-shRNA 3A-sub E#61 (51.90%, *p* = 0.0061) compared with *SIK3*-shRNA 3A-sub E#Luc (23.42%). Left side of red dotted line refers control staining (IgG only), right side as CD204(+) staining. (**D**) The phagocytic ability of THP-1 cells after the coculture with *SIK3*-shRNA 3A-sub E was evaluated using fluorescently prelabeled *E. coli* particles. The phagocytic ability of THP-1 cells after the coculture with *SIK3*-shRNA 3A-sub E#01 (31.05%, *p* = 0.0022) or *SIK3*-shRNA 3A-sub E#61 (41.94%, *p* = 0.0176) was substantially reduced compared with that after the coculture with *SIK3*-shRNA 3A-sub E#Luc (57.55%). Cytochalasin D treatment served as a negative control. * *p* < 0.05; ** *p* < 0.01; *** *p* < 0.001.

**Figure 3 ijms-25-00222-f003:**
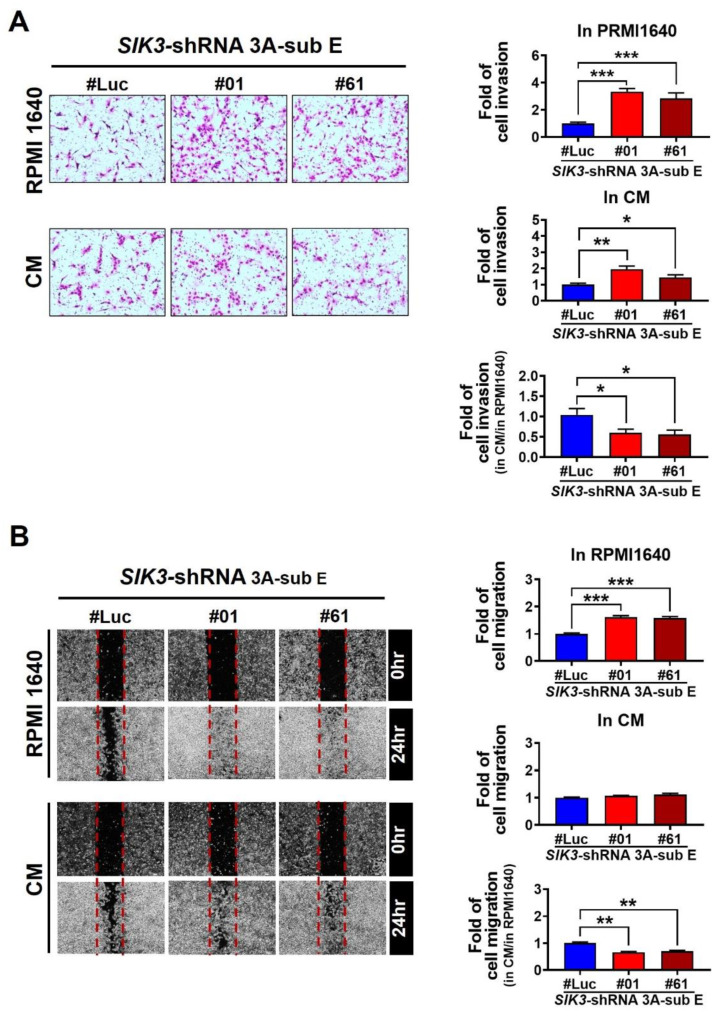
The invasion/migration ability of trophoblast cells decreased in the placental environment with PE. (**A**,**B**) The cell invasion/migration ability of *SIK3*-shRNA 3A-sub E#01 and *SIK3*-shRNA 3A-sub E#61 increased (upper pattern) under the PRMI1640 medium exposure compared with that of *SIK3*-shRNA 3A-sub E#Luc; however, this phenomenon was significantly inhibited by the condition medium (CM, coculture of *SIK3*-shRNA 3A-sub E and THP-1 cells for 7 days) treatment. * *p* < 0.05; ** *p* < 0.01; *** *p* < 0.001.

**Figure 4 ijms-25-00222-f004:**
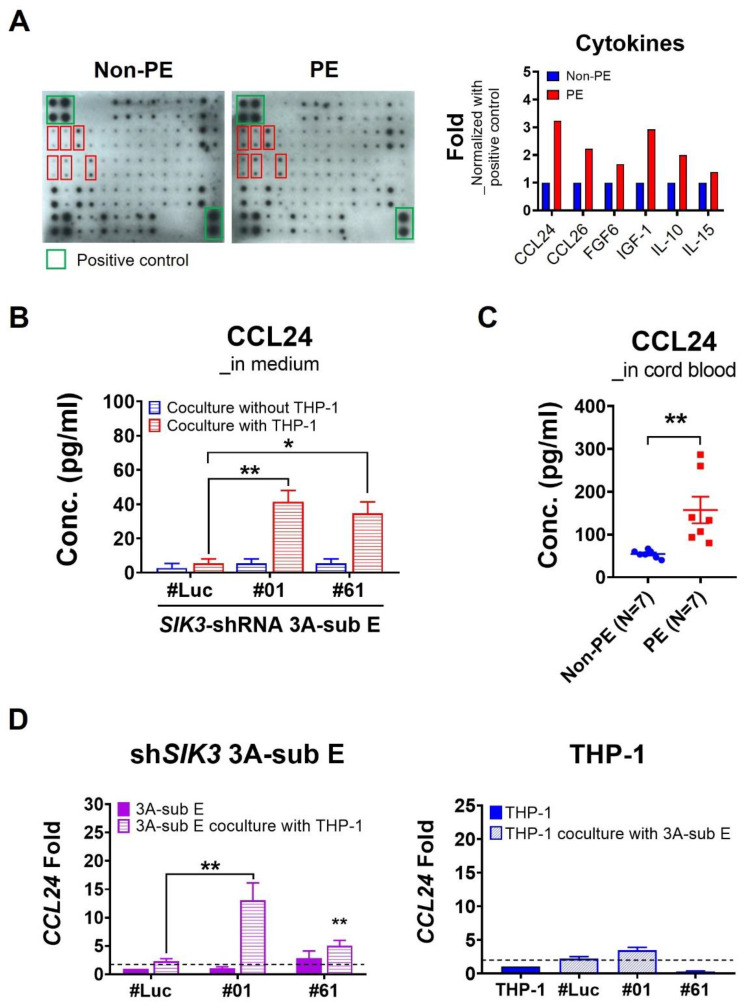
Trophoblast cell-derived CCL24 was abundantly increased in the placental environment with PE. (**A**) Cytokine array patterns showed the differences between non-PE and PE. Six cytokines, namely CCL24, CCL26, FGF6, IGF-1, IL-10, and IL-15 increased in the PE cord blood (a plasma mixture of three PE cases) compared with those in the non-PE (a plasma mixture of three non-PE cases). The red boxes are in order from left to right and from top to bottom: CCL24, CCL26, FGF6, IGF-1, CCL-10 and IL-15. (**B**) In CCL24 ELISA, the CCL24 concentration in the condition medium from *SIK3*-shRNA 3A-sub E#01 (41.33 ± 6.67 pg/mL; *p* = 0.0074) or *SIK3*-shRNA 3A-sub E#61 (34.67 ± 11.55 pg/mL; *p* = 0.0150) after the respective coculture with THP-1 cells for 7 days was higher than that from *SIK3*-shRNA 3A-sub E#Luc cocultured with THP-1 (5.33 ± 2.67 pg/mL). (**C**) Similarly, the CCL24 concentration significantly increased in the cord blood from seven PE cases (157.14 ± 31.16 pg/mL) compared with that from seven non-PE cases (54.29 ± 3.39 pg/mL) with *p* = 0.0066. (**D**) qPCR results showed that the *CCL24* mRNA level significantly increased in *SIK3*-shRNA 3A-sub E#01 (*p* = 0.0078) and *SIK3*-shRNA 3A-sub E#61 (*p* = 0.0072) after the respective coculture with THP-1 cells for 7 days. * *p* < 0.05; ** *p* < 0.01.

**Figure 5 ijms-25-00222-f005:**
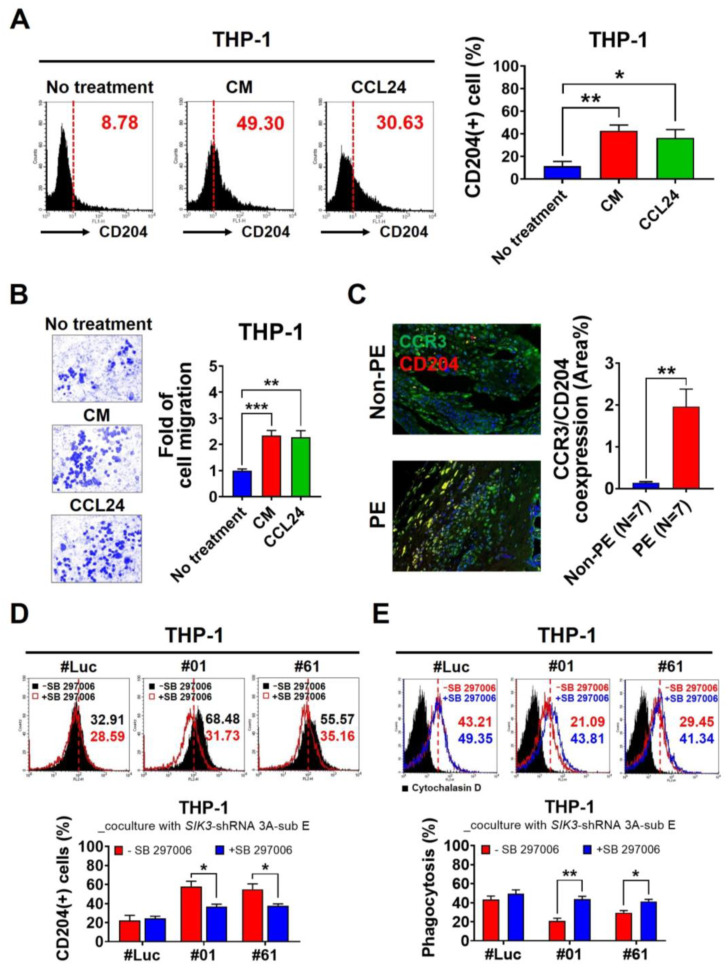
CCL24/CCR3 axis plays a crucial role in the skewing of the M2 macrophage in the placental environment of PE. (**A**) CD204 expression significantly increased in THP-1 cells after CM (from *SIK3*-shRNA 3A-sub E#01 coculture with THP-1 for 7 days, 42.75 ± 4.96%, *p* = 0.0081) or recombinant CCL24 treatment (0.05 ng/mL treatment, mean = 36.32 ± 7.40%, *p* = 0.0419) compared with that in the untreated cells (11.39 ± 4.07%). (**B**) Chemoattractant results showed that the fold of the migrated THP-1 cells significantly increased after recombinant CCL24 (0.05 μg/mL, 2.27 ± 0.25 fold, *p* = 0.0028) and CM (2.30 ± 0.20 fold, *p* = 0.0006) treatment compared that of the untreated cells (1 fold). (**C**) After immunofluorescence staining, the CCR3 and CD204 co-expression at the maternal–fetal interface of the placenta with PE (N = 7) was significantly higher than that without PE (N = 7, *p* = 0.0011). (**D**) Under 2.5 μM SB 297006 exposure, the CD204(+) cells were significantly reduced after the THP-1 coculture with *SIK3*-shRNA 3A-sub E#01 (57.98 ± 5.42% reduced to 36.78 ± 2.57%, *p* = 0.0241) or with *SIK3*-shRNA 3A-sub E#61 (54.96 ± 5.69% reduced to 37.77 ± 1.83%, *p* = 0.0452). (**E**) Under 2.5 μM SB 297006 exposure, the phagocytosis effect of THP-1 cells were significantly increased after coculture with SIK3-shRNA 3A-sub E#01 (21.09 ± 2.61 increased to 43.81 ± 2.88, *p* = 0.0043) or with SIK3-shRNA 3A-sub E#61 (29.45 ± 2.30 increase to 41.34 ± 2.16, *p* = 0.0197). * *p* < 0.05; ** *p* < 0.01; *** *p* < 0.001.

**Figure 6 ijms-25-00222-f006:**
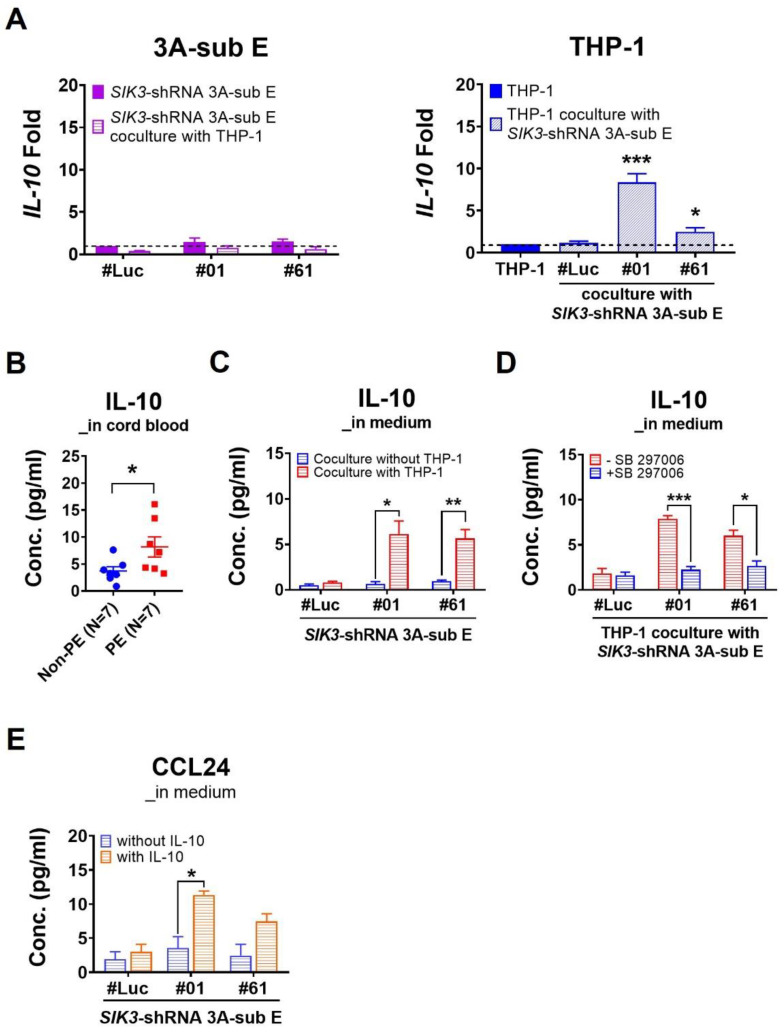
M2 macrophage-derived IL-10 plays a dual role in the placental environment of PE. (**A**) qPCR results demonstrated that the *IL-10* mRNA level substantially increased in THP-1 cells after the coculture with *SIK3*-shRNA 3A-sub E#01 (8.37 ± 1.03 fold, *p* = 0.0004) or *SIK3*-shRNA 3A-sub E#61 (2.49 ± 0.48 fold, *p* = 0.0208). The dashed lines as baseline of IL-10 expression in *SIK3*-shRNA 3A-sub E cells. (**B**,**C**) Through ELISA quantification, the IL-10 concentration in the cord blood with PE (n = 7, 8.17 ± 1.87 pg/mL) significantly increased compare to non-PE (*n* = 7, 3.70 ± 0.80 pg/mL; *p* = 0.0484). Similarly, the IL-10 concentration of the CM significantly increased in *SIK3*-shRNA 3A-sub E#01 (6.13 ± 1.45 pg/mL, *p* = 0.0214) and *SIK3*-shRNA 3A-sub E#61 (5.65 ± 0.98 pg/mL, *p* = 0.0080) after the coculture with THP-1 cells. (**D**) Upon exposure to 2.5 μM CCR3 antagonist (SB 297006), the IL-10 concentration significantly reduced in the CM of THP-1 cells cocultured with *SIK3*-shRNA 3A-sub E#01 (7.88 ± 0.36 pg/mL reduced to 2.25 ± 0.36 pg/mL, *p* = 0.0004) or *SIK3*-shRNA 3A-sub E#61 (6.00 ± 0.63 pg/mL reduced to 2.67 ± 0.55 pg/mL, *p* = 0.0161). (**E**) The CCL24 concentration in the medium significantly increased in *SIK3*-shRNA 3A-sub E#01 (3.56 ± 1.11 pg/mL increased to 11.33 ± 1.11 pg/mL, *p* = 0.0474) after the treatment with 8 pg/mL recombinant IL-10. (**F**) Without recombinant IL-10 treatment, the cell invasion ability of *SIK3*-shRNA 3A-sub E #01 (2.22-fold, *p* = 0.0246) and *SIK3*-shRNA 3A-sub E #61 (1.86-fold, *p* = 0.0079) significantly increased (upper pattern) compared with that of *SIK3*-shRNA 3A-sub E #Luc (1-fold). However, this phenomenon could be inhibited by 8 pg/mL IL-10 (lower pattern). The cell migration of *SIK3*-shRNA 3A-sub E in the treatment with/without IL-10 did not significantly change (**G**). * *p* < 0.05; ** *p* < 0.01; *** *p* < 0.001.

**Figure 7 ijms-25-00222-f007:**
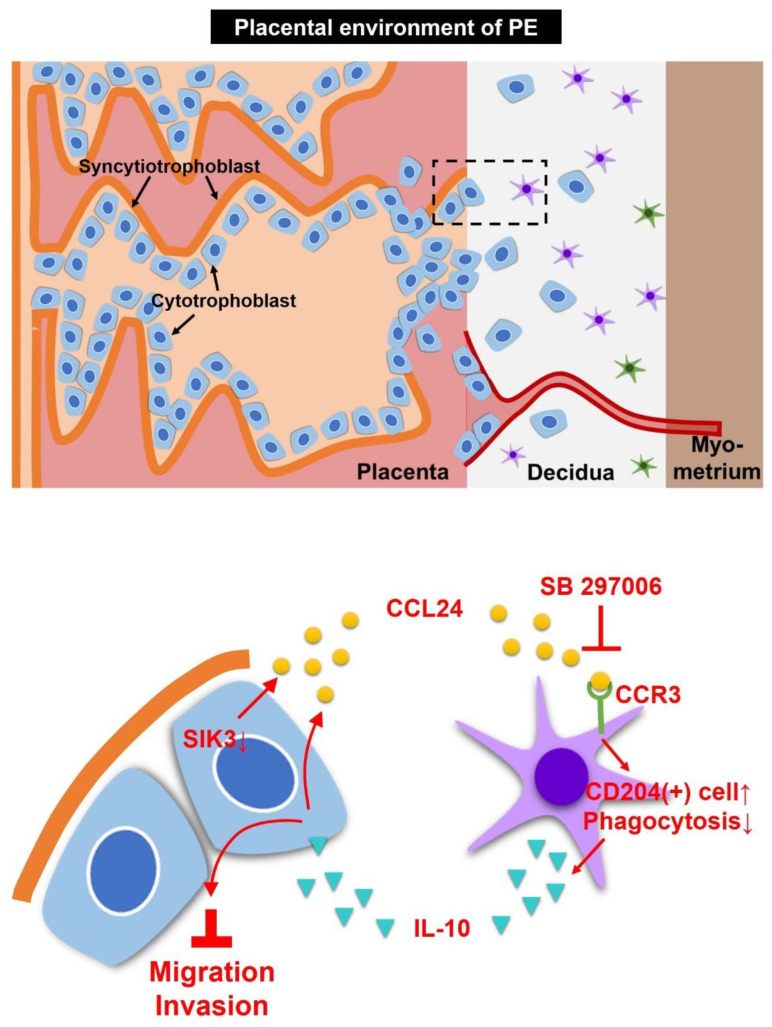
Schematic of the proposed model for the placental environments with PE. SIK3 downregulation in trophoblast cells could promote the secretion of CCL24, leading to the phenotypic skewing of M2 macrophages in the decidua. These increasing CD204(+) cells abundantly released IL-10, reducing the invasion and migration abilities of trophoblast cells. Meanwhile, IL-10 influenced trophoblast cells to produce more CCL24 and maintain M2 macrophage dominance in the placental environments of PE. However, inhibiting the interaction between CCL24 and CCR3 could reverse the M2 dominance in the placental environment of PE.

**Table 1 ijms-25-00222-t001:** Comprehensive analysis of the *SIK3* gene expression dataset in normal pregnancy that is non-PE (non-PE) and preeclamptic (PE).

GEO Dataset	Case Number	*SIK3* Gene Expression (PE/Np)	*p* Value
GSE4707	Non-PE = 4, PE = 9	−0.41	**0.0122**
GSE10588	Non-PE = 26, PE = 5	1.1	0.2893
GSE14722	Non-PE = 11, PE = 12	0.98	0.4915
GSE24129	Non-PE = 8, PE = 8	0.98	**0.0446**
GSE30186	Non-PE = 6, PE = 6	0.73	**0.0164**
GSE43942	Non-PE = 7, PE = 5	1.2	0.3526
GSE54618	Non-PE = 11, PE = 10	−1.17	**0.0253**

**Table 2 ijms-25-00222-t002:** Clinical characteristics, SIK3 expression, CD68 expression, and CD204 expression at the maternal–fetal interface of the placenta with non-preeclampsia and preeclampsia.

	Non-Preeclampsia,	Preeclampsia,	*p* Value
	N = 7 (%)	N = 7 (%)	
**Maternal age (years)**	34 ± 9	34 ± 4	0.28
**Gestational age (weeks)**	39.4 ± 0.7	36.2 ± 2.4	**0.018**
**Newborn’s weight (g)**	3129.3 ± 244.0	2257.7 ± 698.2	**0.005**
**SIK3**			**0.008**
High expression	6 (85.7)	1 (14.3)	
Low expression	1 (14.3)	6 (85.7)	
**CD68**			0.593
High expression	3 (42.8)	4 (47.2)	
Low expression	4 (57.2)	3 (42.8)	
**CD204**			**0.008**
High expression	1 (14.3)	6 (85.7)	
Low expression	6 (85.7)	1 (14.3)	

## Data Availability

Data is contained within the article.

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
