# Peer review of "Downregulation of Salt-Inducible Kinase 3 Enhances CCL24 Activation in the Placental Environment with Preeclampsia"

_ijms, 2023, doi:10.3390/ijms25010222_

Round 1

Reviewer 1 Report

Comments and Suggestions for Authors

I find this paper a valuable element contributing to our knowledge about preeclampsia. The authors analyse the immunological elements in the pathogenesis of PE.

The weak side of this paper is the rather small group of studied patients. I would expect information on why PE patients were delivered earlier than the control group ( if the elective cs were performed or those patients delivered spontaneously). The authors should use term newborn instead of baby

Author Response

Thank you for your comments. There are two main questions, first is “why PE patients were delivered earlier than the control group?” and our answer as below: The prevalence of PE is around 5-7% and it is not easy to collect so many PE cases with severe features. We’ll keep collecting PE cases and expand our case number in our further study. In the meanwhile, PE would be complicated with maternal syndromes of hypertension, proteinuria, and/or end-organ damage, as well as intrauterine growth restriction and fetal distress. These maternal and fetal complications would increase the possibility of preterm delivery due to no sufficient treatment for PE except for termination of pregnancy. The secondary question is "The authors should use term newborn instead of baby" and our answer as below: We changed the "baby" to "newborn" in this article, and label it in green color in Table 2, line 385 and line 390.

Reviewer 2 Report

Comments and Suggestions for Authors

Dear author

Greetings! The work is interesting  but major corrections are needed

1Down regulation of salt-inducible kinase 3 increases CCL24 ac- 2 tivation in placenta with preeclampsia(Please modify title and change increase word )

2Nonetheless, the widely 14 accepted notion that successful pregnancy relies on maternal immunological adaptation is of utmost 15 importance. Furthermore, salt-inducible kinase 3 (SIK3), an AMP-activated protein kinase-related 16 kinase, is a novel regulator of energy and inflammation. To explore the role of SIK3 in PE, we 17 searched GEO datasets for SIK3 gene expression and its association with PE. We identified that SIK3 18 was significantly downregulated in PE across four datasets (p<0.05), suggesting that SIK3 partici- 19 pated in the pathogenesis of PE. (Please modify and expand the terms )

3This condition results in intrauterine growth restriction (IUGR) and 35 maternal syndromes of hypertension, proteinuria, and/or end-organ damage, which usu- 36 ally manifests after the mid-trimester of pregnancy [1-4]. The combined biomarkers of PE, 37 including cytokines, proteins, angiogenic and antiangiogenic factors, play pivotal roles in 38 the pathogenesis and etiology of PE, as well as in other related hypertensive disorder of 39 pregnancy. However, the exact pathophysiology of PE remains unclear, and early predic- 40 tion of PE act as a significant challenge. 41 A successful pregnancy depends on maternal immunologic tolerance systems to bal- 42 ance the allogeneic fetus and placenta [5]. (Please recheck)

4 actually immune reaction or modulation or auto immune please confirm and rewrite (In other words, the detected levels of TNF-α or IL-6 54 in circulation cannot serve as reliable predicable markers of PE. To date, increasing evi- 55 dence suggests that an imbalance in immunologic modulation at the maternal–fetal inter- 56 face of the placenta contributes to certain pregnancy complications, such as PE and IUGR 57 [11]. Given the dynamic function of SIK3 in immune modulation)

6table -1 and figure 1 please allign as per journal guide lines PE compared with non-PE. We then used a CCL24 ELISA kit to validate the CCL24 con- 169 centration in the CM and in 14 cord blood samples with/without PE. ELISA data showed 170 a significant increase in CCL24 in the CM of SIK3-shRNA 3A-sub E#01 (mean: 41.33 pg/ml, 171 p=0.0074) and SIK3-shRNA 3A-sub E#61 (mean: 34.67 pg/ml, p=0.0150) after the coculture 172 with THP-1 cells for 7 days (Figure 4B). Similarly, the CCL24 concentration in the cord 173 blood samples with PE was higher than that in the non-PE samples (mean: 157.10 pg/ml 174 vs 54.29 pg/ml, p=0.0066; Figure 4C).

8To clarify the pivotal role of CCL24, we used SB 297006, a CCR3 antagonist, to assess 209 whether the CCL24/CCR3 axis is the major contributor to M2 polarization in the placental 210 environments with PE. We observed that the CD204(+) cell population significantly de- 211 creased in THP-1 cells after the coculture with SIK3-shRNA 3A-sub E#01 (57.98±5.42% 212 reduction to 36.78±2.57%, p=0.0241) and 2.5 μM SB 297006 compared with that of non-SB 213 297006 exposure. The same result was observed after the coculture with SIK3-shRNA 3A- 214 sub E#61 (54.96±5.69% reduction to 37.77±1.83%, p=0.0452) and 2.5 μM SB 297006 (Figure 215 5D). Additionally, SB 297006 exposure reversed the phagocytotic effect on THP-1 cells 216 after the coculture with SIK3-shRNA 3A-sub E#01 (21.09±2.61 increased to 43.81±2.88, 217 p=0.0043) or SIK3-shRNA 3A-sub E#61 (29.45±2.30 increase to 41.34±2.16, p=0.0197; Figure 218 5E). These results suggested that trophoblast cell-derived CCL24 sufficiently altered the 219 phenotype and function of maternal macrophages in the placental environments with PE 220 by interacting with CCR3. 2 Please alter the sentences 

9CM of THP-1 cocultured with SIK3-shRNA 3A-sub E#61 (mean: 6.00 250 pg/ml reduced to 2.67, p=0.0161) upon 2.5 μM SB 297006 exposure(Please modify and rewrite)

Interestingly, we noticed an increase in the IL-10 level in the cord blood of PE through 346 cytokine array analysis (Figure 4A), while IL-10 has been demonstrated as a key anti-in- 347 flammatory mediator to regulate wound healing, autoimmunity, cancer, and homeostasis 348 [28]. The dysregulation of IL-10 is associated with the occurrence of PE although the IL- 349 10 detection yields inconsistent results Its not clear 

Phagocytosis figure should be included to confirm

Please allign references by using zotero or mendelley 

Best regards

Comments on the Quality of English Language

English language fine but they have to expand the terms

Author Response

Response to reviewer 2

Question 1: Down regulation of salt-inducible kinase 3 increases CCL24 activation in placenta with preeclampsia (Please modify title and change increase word)

Answer: Thank you for your advice. We change the title “Down regulation of salt-inducible kinase 3 increases CCL24 activation in placenta with preeclampsia” to “Down regulation of salt-inducible kinase 3 enhances CCL24 activation in the placental environment with preeclampsia”. The modifications are label in red color.

Question 2: Nonetheless, the widely accepted notion that successful pregnancy relies on maternal immunological adaptation is of utmost importance. Furthermore, salt-inducible kinase 3 (SIK3), an AMP-activated protein kinase-related kinase, is a novel regulator of energy and inflammation. To explore the role of SIK3 in PE, we searched GEO datasets for SIK3 gene expression and its association with PE. We identified that SIK3 was significantly downregulated in PE across four datasets (p<0.05), suggesting that SIK3 participated in the pathogenesis of PE. (Please modify and expand the terms)

Answer: Thank you for your comment. We change the sentences to "The recent widely accepted notion that successful pregnancy relies on maternal immunological adaptation is of utmost importance. Moreover, salt-inducible kinase 3 (SIK3) that is an AMP-activated protein kinase-related kinase and it has reported a novel regulator of energy and inflammation. To explore the SIK3 expression whether correlated with PE, we analyzed SIK3 gene expression and its association with PE through GEO datasets. We identified that SIK3 was significantly downregulated in PE across four datasets (p<0.05), suggesting that SIK3 participated in the pathogenesis of PE." on line 14-20 and label in red color.

Question 3: This condition results in intrauterine growth restriction (IUGR) and maternal syndromes of hypertension, proteinuria, and/or end-organ damage, which usually manifests after the mid-trimester of pregnancy [1-4]. The combined biomarkers of PE, including cytokines, proteins, angiogenic and antiangiogenic factors, play pivotal roles in the pathogenesis and etiology of PE, as well as in other related hypertensive disorder of pregnancy. However, the exact pathophysiology of PE remains unclear, and early prediction of PE act as a significant challenge. A successful pregnancy depends on maternal immunologic tolerance systems to balance the allogeneic fetus and placenta [5]. (Please recheck)

Answer: Thank you for your advice. We delete “pathogenesis and etiology of PE, as well as in other” on line 39. And we change " A successful pregnancy depends on maternal immunologic tolerance systems to balance the allogeneic fetus and placenta [5]" to " A successful pregnancy depends on maternal immunologic tolerance systems to balance the allogeneic fetus and placenta is wildly accepted now [5]." on line 43. The modifications are label in red color.

Question 4: actually immune reaction or modulation or auto immune please confirm and rewrite (In other words, the detected levels of TNF-α or IL-6 in circulation cannot serve as reliable predicable markers of PE. To date, increasing evidence suggests that an imbalance in immunologic modulation at the maternal–fetal interface of the placenta contributes to certain pregnancy complications, such as PE and IUGR [11]. Given the dynamic function of SIK3 in immune modulation)

Answer: Thank you for your comment. Indeed, SIK3 affects the immune response of the environment by regulating the macrophages differentiation. So we changed “immune modulation” to “immune reaction”, and label it in red color in line 58.

Question 5: table -1 and figure 1 please allign as per journal guide lines PE compared with non-PE.

Answer: Thank you for your comment. To avoid confusion, we change the "Np" in Table 1 to "Non-PE", even though they refer to the same population. Label it in red color in line 90-91 and Table 1.

Question 6: We then used a CCL24 ELISA kit to validate the CCL24 concentration in the CM and in 14 cord blood samples with/without PE. ELISA data showed a significant increase in CCL24 in the CM of SIK3-shRNA 3A-sub E#01 (mean: 41.33 pg/ml, p=0.0074) and SIK3-shRNA 3A-sub E#61 (mean: 34.67 pg/ml, p=0.0150) after the coculture with THP-1 cells for 7 days (Figure 4B). Similarly, the CCL24 concentration in the cord blood samples with PE was higher than that in the non-PE samples (mean: 157.10 pg/ml vs 54.29 pg/ml, p=0.0066; Figure 4C).

Answer: We re-modify this text as " After the coculture with THP-1 cells for 7 days, the CCL24 concentration increased significantly in the CM of SIK3-shRNA 3A-sub E#01 (41.33±6.67 pg/ml, p=0.0074) and SIK3-shRNA 3A-sub E#61 (34.67±11.55 pg/ml, p=0.0150) compared with the CM of SIK3-shRNA 3A-sub E#Luc (5.33±2.67 pg/ml), respectively (Figure 4B, red histogram). Conversely, the CCL24 concentration revealed no significant change in the CM of SIK3-shRNA 3A-sub E without the coculture with THP-1 cells (Figure 4B, blue histogram). Similarly, the CCL24 concentration in the cord blood samples with PE was higher than that in the non-PE samples (157.14±31.16 pg/ml vs 54.29±3.39 pg/ml, p=0.0066; Figure 4C)." Label it in red color in the line 171-179.

Question 7: To clarify the pivotal role of CCL24, we used SB 297006, a CCR3 antagonist, to assess whether the CCL24/CCR3 axis is the major contributor to M2 polarization in the placental environments with PE. We observed that the CD204(+) cell population significantly decreased in THP-1 cells after the coculture with SIK3-shRNA 3A-sub E#01 (57.98±5.42% reduction to 36.78±2.57%, p=0.0241) and 2.5 μM SB 297006 compared with that of non-SB 297006 exposure. The same result was observed after the coculture with SIK3-shRNA 3A-sub E#61 (54.96±5.69% reduction to 37.77±1.83%, p=0.0452) and 2.5 μM SB 297006 (Figure 5D). Additionally, SB 297006 exposure reversed the phagocytotic effect on THP-1 cells after the coculture with SIK3-shRNA 3A-sub E#01 (21.09±2.61 increased to 43.81±2.88, p=0.0043) or SIK3-shRNA 3A-sub E#61 (29.45±2.30 increase to 41.34±2.16, p=0.0197; Figure 5E). These results suggested that trophoblast cell-derived CCL24 sufficiently altered the phenotype and function of maternal macrophages in the placental environments with PE by interacting with CCR3. Please alter the sentences 

Answer: Thank you for your comment. We alter the sentences to "We further used SB 297006, a CCR3 antagonist, to validate the CCL24/CCR3 axis whether is the major contributor to M2 polarization in the placental environments with PE. We observed the CD204(+) cell population decreased significantly in THP-1 cells after the coculture with SIK3-shRNA 3A-sub E#01 and treated with 2.5 μM SB 297006 (57.98±5.42% reduction to 36.78±2.57%, p=0.0241). The same result was observed after the coculture with SIK3-shRNA 3A-sub E#61 and treated with 2.5 μM SB 297006 (54.96±5.69% reduction to 37.77±1.83%, p=0.0452) (Figure 5D). Additionally, the SB 297006 exposure could reverse the phagocytotic effect on THP-1 cells, regardless after the coculture with SIK3-shRNA 3A-sub E#01 (21.09±2.61 increased to 43.81±2.88, p=0.0043) or SIK3-shRNA 3A-sub E#61 (29.45±2.30 increase to 41.34±2.16, p=0.0197) (Figure 5E). These results suggested that trophoblast cell-derived CCL24 sufficiently altered the phenotype and function of maternal macrophages in the placental environments with PE by interacting with CCR3." Label it in red color in the line 213-225.

Question 8: CM of THP-1 cocultured with SIK3-shRNA 3A-sub E#61 (mean: 6.00 pg/ml reduced to 2.67, p=0.0161) upon 2.5 μM SB 297006 exposure (Please modify and rewrite)

Answer: We alter the sentences to " In the CM of THP-1 cells cocultured with SIK3-shRNA 3A-sub E#01, the level of IL-10 could be significantly reduced upon 2.5 μM SB 297006 exposure (7.88±0.36 pg/ml reduced to 2.25±0.36 pg/ml, p=0.0004). The same result can be observed in the CM of THP-1 cells cocultured with SIK3-shRNA 3A-sub E#61 after 2.5 μM SB 297006 treatment (6.00±0.63 pg/ml reduced to 2.67±0.55 pg/ml, p=0.0161) (Figure 6D)" Label it in red color in the line 252-256.

Question 9: Interestingly, we noticed an increase in the IL-10 level in the cord blood of PE through cytokine array analysis (Figure 4A), while IL-10 has been demonstrated as a key anti-inflammatory mediator to regulate wound healing, autoimmunity, cancer, and homeostasis [28]. The dysregulation of IL-10 is associated with the occurrence of PE although the IL- 10 detection yields inconsistent results. Its not clear. Phagocytosis figure should be included to confirm

Answer: Thank you for your advice. IL-10 has reported plays a vital role in maintaining the balance of anti-inflammatory and pro-inflammatory milieu at the maternal–fetal interface (J Reprod Immunol. 2011;88(2):165-169). However, we demonstrated the source of IL-10 increasing in placental microenvironment is from macrophage (Figure 6A), and the macrophage phenotype trends toward M2 in this statu (Figure 2C). When we used the SB 297006 treatment, the M2 phenotype obviously decreased (Figure 5D) and the IL-10 level were significantly decreased (Figure 6D). The results imply IL-10 is the product of M2 rather than regulator to change the function of macrophage. We used flow cytometry to analyze phagocytic effect of THP-1 cells rather than fluorescence image, that is because we can get the analyze data directly.

Question 10: Please allign references by using zotero or mendelley 

Answer: Thank you for your advice. We were follow the format required by the IJMS to edit references. We also know that some software is very convenient, such as EndNote. The reason of we edit the reference by ourselves is we also perform double confirm of the citation. In addition, we need time to be familiar with the operation of zotero software, this may not be accomplished in a few days.
